# Helper NLR immune protein NRC3 evolved to evade inhibition by a cyst nematode virulence effector

Yu Sugihara[1], Jiorgos Kourelis[1,¤], Mauricio P. Contreras[1], Hsuan Pai[1], Adeline Harant[1], Muniyandi Selvaraj[1], AmirAli Toghani[1], Claudia Martínez-Anaya[1,2*], Sophien Kamoun[1*]

1 The Sainsbury Laboratory, University of East Anglia, Norwich, United Kingdom, 2 Instituto de Biotecnología, Universidad Nacional Autónoma de México, Cuernavaca, Morelos, México

¤ Current address: Department of Life Sciences, Imperial College London, London, United Kingdom
* sophien.kamoun@tsl.ac.uk (SK); claudia.martinez@ibt.unam.mx (CM-A)

## Abstract

Parasites can counteract host immunity by suppressing nucleotide binding and leucine-rich repeat (NLR) proteins that function as immune receptors. We previously showed that a cyst nematode virulence effector SPRYSEC15 (SS15) binds and inhibits oligomerisation of helper NLR proteins in the expanded NRC1/2/3 clade by preventing intramolecular rearrangements required for NRC oligomerisation into an activated resistosome. Here we examined the degree to which NRC proteins from multiple Solanaceae species are sensitive to suppression by SS15 and tested hypotheses about adaptive evolution of the binding interface between the SS15 inhibitor and NRC proteins. Whereas all tested orthologs of NRC2 were inhibited by SS15, some natural variants of NRC1 and NRC3 are insensitive to SS15 suppression. Ancestral sequence reconstruction combined with functional assays revealed that NRC3 transitioned from an ancestral suppressed form to an insensitive one over 19 million years ago. Our analyses revealed the evolutionary trajectory of an NLR immune receptor against a parasite inhibitor, identifying key evolutionary transitions in helper NLRs that counteract this inhibition. This work reveals a distinct type of gene-for-gene interaction between parasite or pathogen immunosuppressors and host immune receptors that contrasts with the coevolution between AVR effectors and immune receptors.

## Author summary

Plants have immune systems that protect them from various pathogens, including bacteria, fungi and nematodes. This immune system harnesses proteins called NLRs (nucleotide-binding and leucine-rich repeat proteins) to detect and respond to pathogens. However, pathogens often evolve strategies to suppress plant immunity. For example, the cyst nematode *Globodera rostochiensis* produces an effector protein called SPRYSEC15 (SS15) that inhibits certain NLR proteins, suppressing the plant's immune response. In this study, we explored how a group of NLR proteins called NRCs, found in

**Data availability statement:** The datasets used in this study are archived in Zenodo (https://doi.org/10.5281/zenodo.14584438). The ancestral sequence reconstruction pipeline "anceseq" is available on Github (https://github.com/YuSugihara/anceseq) and archived in Zenodo (https://doi.org/10.5281/zenodo.10808871).

**Funding:** 1. This study was supported by The Gatsby Charitable Foundation (YS, JK, MPC, HP, AH, MS, AT, CMA, SK), Biotechnology and Biological Sciences Research Council (BBSRC) BB/P012574 (Plant Health ISP) (YS, JK, MPC, HP, AH, MS, AT, CMA, SK), BBSRC BBS/E/J/000PR9795 (Plant Health ISP - Recognition) (YS, JK, MPC, HP, AH, MS, AT, CMA, SK), BBSRC BBS/E/J/000PR9796 (Plant Health ISP - Response) (YS, JK, MPC, HP, AH, MS, AT, CMA, SK), BBSRC BBS/E/J/000PR9797 (Plant Health ISP – Susceptibility) (YS, JK, MPC, HP, AH, MS, AT, CMA, SK), BBSRC BBS/E/J/000PR9798 (Plant Health ISP – Evolution) (YS, JK, MPC, HP, AH, MS, AT, CMA, SK), BBSRC BB/V002937/1 (SK, MPC), European Research Council (ERC) 743165 (SK). More information about the funding sources can be found at the following websites: the Gatsby Charitable Foundation (https://www.gatsby.org.uk/), BBSRC (https://www.ukri.org/councils/bbsrc/), ERC (https://erc.europa.eu) The funders had no role in the study design, data collection and analysis, decision to publish, or preparation of the manuscript.

**Competing interests:** I have read the journal's policy and the authors of this manuscript have the following competing interests: S.K. receives funding from industry on NLR biology and has cofounded a start-up company (Resurrect Bio Ltd.) related to NLR biology. J.K., M.P.C. and S.K. have filed patents on NLR biology. M.P.C. has received fees from Resurrect Bio Ltd.

the Solanaceae plant family, have evolved to counteract SS15 inhibition. We discovered that while SS15 inhibits all tested NRC2 orthologs, some natural variants of NRC1 and NRC3 are insensitive to its effects. By reconstructing the evolutionary history of NRC3, we found that these proteins acquired insensitivity to SS15 through a specific amino acid substitution about 19 million years ago. Our findings highlight a previously understudied type of evolutionary arms race between plants and pathogens, where plants adapt to evade pathogen-derived suppressors. This research expands our understanding of plant-pathogen coevolution and offers insights that could guide the development of crops with improved disease resistance.

## Introduction

Molecular interactions between plant immune receptors and pathogen effectors drive their coevolution, shaping the dynamics of host-pathogen relationships [1–4]. Harold H. Flor, in 1942, proposed the hypothesis that single genes in plants and pathogens determine the outcome of their interactions; that is, a plant carrying a certain gene displays resistance against a pathogen carrying a corresponding gene [5,6]. In plant-pathogen interactions, Flor's model is generally interpreted as the recognition of pathogen effectors (known as AVR effectors) by plant immune receptors (encoded by disease resistance genes or *R* genes), triggering a robust resistance mechanism. This gene-for-gene model has provided an invaluable conceptual framework and has significantly influenced both applied and basic research in disease resistance [7]. In particular, this model provided a deep understanding of the principles of coevolution between plant immune receptors, encoded by disease resistance (*R*) genes, and pathogen (or parasite) AVR effectors [8–11]. Indeed, the plant pathology community has documented numerous examples of adaptive evolution, notably cases where the pathogen AVR effector has evaded detection by its cognate R immune receptor [11–13]. Despite advancements in our understanding of R-AVR interactions at the mechanistic and evolutionary level, other types of gene-for-gene interactions, notably coevolution between pathogen immunosuppressors and their host targets, remain less understood [14,15] (S1 Fig).

Nucleotide-binding and leucine-rich repeat (NLR) proteins constitute the predominant class of plant disease resistance genes [16–18]. NLRs are intracellular receptors that directly or indirectly recognise pathogen effectors, a molecular event which triggers an immune response [18,19]. A continuous coevolutionary arms race between plants and pathogens has resulted in the diversification and expansion of NLRs in plants [14,20,21]. NLRs have evolved diverse activation mechanisms. Asterid plants, and notably the Solanaceae, have a complex NLR immune receptor network also known as the NRC network [14,16,22]. In this model, a specific phylogenetic clade of coiled-coil (CC)-type NLR proteins, known as NLR REQUIRED FOR CELL DEATH (NRC), function as helper NLRs (NRC-H) genetically downstream of multiple sensor NLRs (NRC-S) to mediate immune responses and to confer disease resistance against diverse pathogens and pests [22–24].

NRC-H and NRC-S are phylogenetically related CC-NLRs that originate from a common ancestor that was present before the split between asterids and Caryophyllales [22]. Recent studies have shown that the NRC0 sensor/helper gene cluster reflects an ancestral state of the NRC network [23,24]. *NRC0* helpers have been genetically and functionally conserved across asterids for more than 100 million years, predating the massive expansion and diversification of NRC-H and NRC-S into lineage- or species-specific NRC networks in lamiids, notably in the Solanaceae family [23–25]. Throughout their expansion and diversification, NRC proteins

have subfunctionalised from the larger NRC network, resulting in organ-specialised subnetworks or in partial genetic redundancy as well as evolving specificity to different sensor NLRs [26,27]. Notably, NRC3 of solanaceous species is required not only by sensor NLRs but also by the tomato cell-surface receptors Cf-2, Cf-4, Cf-5 and Cf-9, which trigger hypersensitive cell death response upon the recognition of the fungal pathogen *Cladosporium fulvum* (syn. *Passalora fulva*) apoplastic effectors [28–30].

Upon perceiving AVR effectors, NRC-S proteins activate their cognate NRC helpers via an 'activate-and-release' mechanism, presumably through the NB-ARC (nucleotide-binding adaptor shared by APAF-1, certain *R* gene products and CED-4) module of the sensor NLR [26,31–33]. The NB-ARC module consists of three domains: a nucleotide binding domain (NBD), a helical domain (HD1) and a winged helix domain (WHD) [34–36]. The current mechanistic model is that activation by NRC-S converts NRC helpers from a resting state homodimer into resistosomes—oligomeric, pore-like complexes that translocate to the plasma membrane [33,37]. These complexes enable immune responses such as calcium influx and hypersensitive cell death, culminating in disease resistance [38]. Oligomerisation into resistosomes has been demonstrated *in planta* for several NRCs using blue native polyacrylamide gel electrophoresis (BN-PAGE) [24,31–33,35]. More recently, the cryogenic electron microscopy (cryo-EM) structures of sensor-activated NRC2 and a constitutively active mutant of NRC4 (NRC4$^{D478V}$) revealed hexameric resistosomes [38,39].

We previously showed that SPRYSEC15 (SS15), a small secreted protein of the cyst nematode *Globodera rostochiensis*, binds and inhibits oligomerisation of NRC2 and NRC3 from the model plant *Nicotiana benthamiana*, as well as NRC1 from tomato (*Solanum lycopersicum*), by preventing intramolecular rearrangements required for NRC oligomerisation into activated resistosomes [35,40,41]. Conversely, SS15 does not inhibit oligomerisation of *N. benthamiana* NRC4 protein, a paralog of NRC2 and NRC3 [35]. Leveraging NRC4 resilience to SS15 inhibition and the crystal structure of SS15 in complex with the NB-ARC module of NRC1, Contreras et al. (2023) mapped two key residues, E316 and D317, on NRC2 that mediate binding to SS15 and showed that NRC2 variants mutated at these positions can evade SS15 suppression without compromising receptor signalling [35]. However, these studies focused mainly on NRC2, NRC3, and NRC4 from *N. benthamiana*, without examining the sensitivity of NRC proteins across Solanaceae to SS15 inhibition. Indeed, Solanaceae species have 7–21 NRC-H paralogs [24], and the phylogenomic analyses of 123 solanaceous genome assemblies from 39 species of the genera *Solanum*, *Physalis*, *Capsicum* and *Nicotiana* revealed 15 phylogenetic clades [37,42]. Ten of the 15 clades include orthologs from at least two genera, therefore representing relatively deep NRC lineages. Of interest to this study, SS15 has only been shown to inhibit members of the NRC1, NRC2, NRC3 clade, a well-supported and deep branch in the Solanaceae NRC tree [37].

The extent to which NLR immune receptors have evolved to counteract pathogen suppressors is unknown. Here we discovered that orthologous NRC proteins from multiple Solanaceae species vary in their sensitivity to suppression by the cyst nematode effector SS15. We found that some natural variants of NRC1 and NRC3 are insensitive to SS15 suppression, whereas all tested NRC2 orthologs are inhibited by SS15. To test hypotheses about adaptive evolution of the NRC-SS15 binding interface, we used ancestral sequence reconstruction to show that NRC3 has transitioned from an ancestral suppressed form to an insensitive one over 19 million years ago. Our analyses reconstructed the evolutionary trajectory of NLRs against their suppressors and identified key evolutionary transitions of solanaceous helper NLRs to counteract the SS15 inhibitor. This work reveals a distinct type of gene-for-gene interaction between host immune receptors and pathogen immunosuppressors that contrasts with R-AVR coevolution.

## Results

### Natural variants of NRC1 and NRC3 but not NRC2 are insensitive to SS15 inhibition

We previously showed that SS15 inhibits NRC1, NRC2 and NRC3 but not NRC4 [35,41]. Given that the homologous NRC1, NRC2, NRC3, along with the NLR modulator NRCX, form a well-supported clade in the Solanaceae NRC tree (Figs 1A and S2) [24,37], we investigated the degree to which NRC orthologs from multiple Solanaceae species are inhibited by SS15. For this purpose, we cloned 11 representative NRC sequences from *Nicotiana benthamiana* (NbNRC2 and NbNRC3), *Capsicum annuum* (pepper; CaNRC1, CaNRC2 and CaNRC3), *Solanum tuberosum* (potato; StNRC1, StNRC2 and StNRC3) and

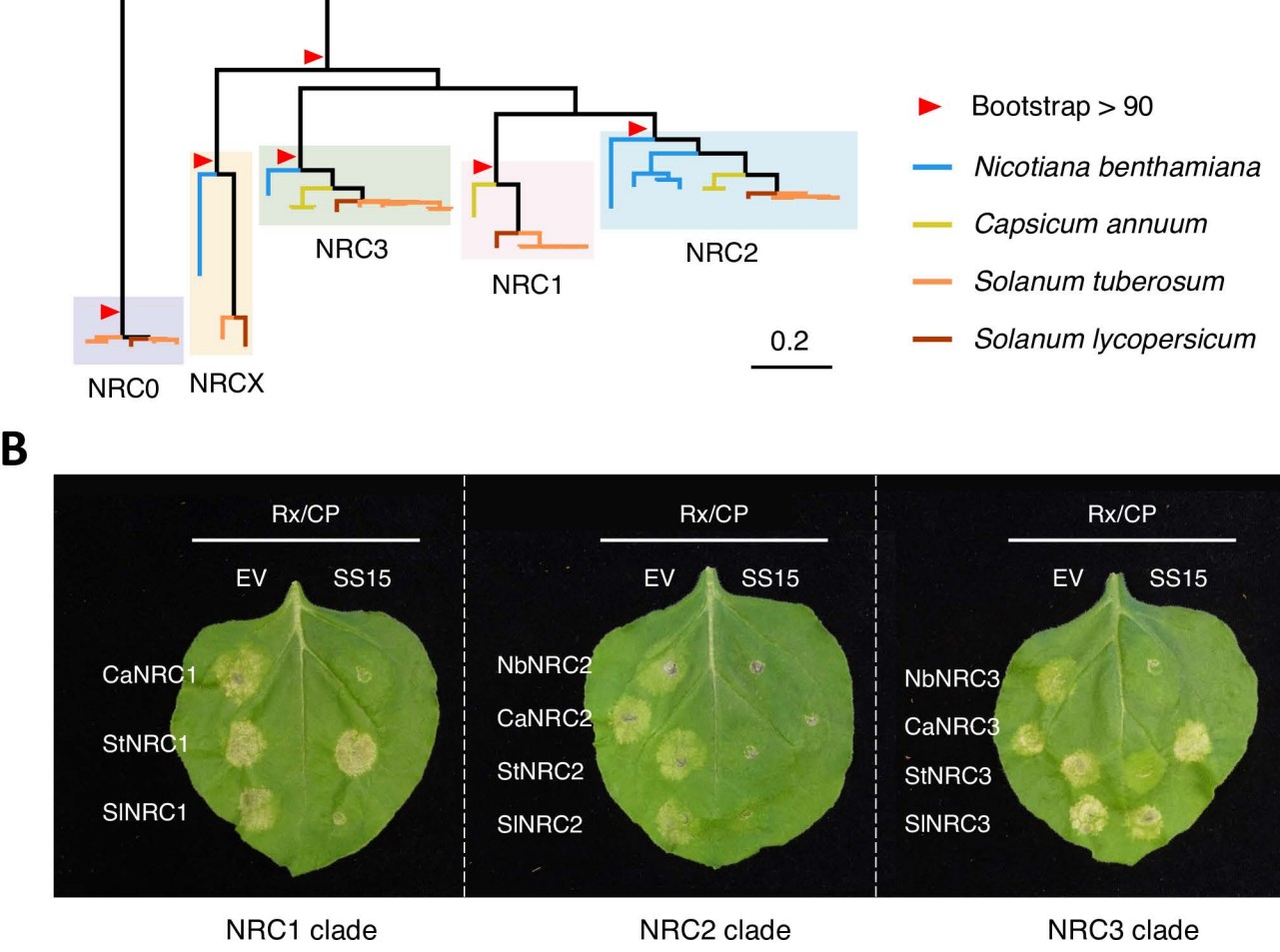

**Fig 1. Natural variants of NRC1 and NRC3 but not NRC2 are insensitive to SS15 inhibition.** (A) Phylogenetic tree of the NRC1/2/3 clade from four different Solanaceae species. We used the full-length NRC sequences of the NRC0, NRC1, NRC2, NRC3 and NRCX clades from four different Solanaceae species (*Nicotiana benthamiana*, *Capsicum annuum*, *Solanum tuberosum* and *Solanum lycopersicum*) to create a phylogenetic tree. The phylogenetic tree was reconstructed by the maximum likelihood method using IQ-TREE with 1,000 bootstrap replicates. The phylogenetic positions of cloned NRCs are indicated in S2 Fig. (B) Representative images of HR cell death assays after transient co-expression of either an empty vector (EV) or SPRYSEC15 (SS15) with the NRC-S sensor NLR, Rx, the *Potato virus X* (PVX) coat protein (CP) and NRCs in the leaves of *N. benthamiana nrc2/3/4* KO plants. Note that, unlike tomato, *N. benthamiana* lacks *nrc1*. Leaves were photographed 5 days after infiltration.

*Solanum lycopersicum* (tomato; SlNRC1, SlNRC2 and SlNRC3) from the phylogenetically related NRC1/2/3 clade (Figs 1 and S2). To assess cell death due to hypersensitive response (HR) in the presence of SS15, we transiently co-expressed NRCs with the upstream NRC-S sensor NLR, Rx, along with either an empty vector (EV) or SS15 in the leaves of *nrc2/3/4* CRISPR knock-out (KO) *N. benthamiana* plants. We activated the sensor-helper Rx-NRC system by co-expressing the *Potato virus X* (PVX) coat protein (CP). Whereas all tested NRC2s were inhibited by SS15, some natural variants of NRC1 and NRC3 were insensitive to SS15 inhibition (Fig 1B). For NRC1, pepper NRC1 (CaNRC1) and tomato NRC1 (SlNRC1) were inhibited by SS15, while potato NRC1 (StNRC1) was insensitive to SS15 inhibition (Fig 1B). For NRC3, *N. benthamiana* NRC3 (NbNRC3) and potato NRC3 (StNRC3) were inhibited by SS15, while pepper NRC3 (CaNRC3) and tomato NRC3 (SlNRC3) were insensitive (Fig 1B). These results indicate that some natural variants of NRC1 and NRC3 are insensitive to SS15 and that they might have acquired mutations affecting SS15 binding to their NB-ARC modules.

## NRC3 variants from different Solanaceae species carry key polymorphisms at the SS15 binding interface

Contreras et al. (2023) described the structure of SS15 in complex with the NB-ARC module of SlNRC1 (SlNRC1^NB-ARC) revealing three contact interfaces (Fig 2A) [35]. To interpret the results of the cell death assay in Fig 1B, we mapped the NRC polymorphisms at these interfaces (Fig 2B). Among multiple polymorphisms, we focused on positions D316 and E317 of SlNRC1, which are critical to SS15 inhibition in NbNRC2 [35]. In contrast to NRC1 and NRC2, NRC3 is polymorphic at positions matching D316 and E317 (here and below, the position numbers are based on SlNRC1 sequence) (Fig 2B). Remarkably, polymorphism E316K in NRC3 correlated with inhibition phenotypes, with CaNRC3 and SlNRC3 orthologs, carrying lysine at position 316 (K316), being insensitive to SS15 inhibition (Figs 1B and 2B). We therefore hypothesised that this amino acid substitution at position matching 316 affects binding and sensitivity to SS15 in the NRC3 clade.

## Polymorphisms at NRC3 position 316 determine sensitivity to SS15 suppression

Given the findings described in Figs 1 and 2 and since NRC3 is a core helper NLR required not only by sensor NLRs but also by cell surface receptors [28–30], we focused on NRC3 and tested the effect of E316K substitution on SS15 binding. We generated a single-point mutant for each wild type (WT) NRC3 at position matching SlNRC1 amino acid 316 (E316K in NbNRC3, K314E in CaNRC3, E314K in StNRC3 and K314E in SlNRC3; Figs 2B and 3A). We then compared the HR cell death between WT NRC3s and single-point mutants in the presence and absence of SS15 (Fig 3B and 3C). The E-to-K substitutions in NbNRC3 and StNRC3 significantly increased the cell death response in the presence of SS15, while the K-to-E substitutions in CaNRC3 and SlNRC3 significantly reduced it (Fig 3B and 3C). The differences are not due to SS15-mediated destabilization of NRCs as there is no correlation between SS15 suppression levels and protein accumulation levels (S3 Fig). SlNRC3^K314E showed statistically reduced HR scores compared with WT SlNRC3 in the presence of SS15, but the difference was not as marked as in the case of other single-point mutants (Figs 3B, 3C and S3). These results indicate that naturally occurring polymorphisms at position 316 of NRC3 are critical for sensitivity to SS15 in line with our previous study on NbNRC2 [35]. Furthermore, our findings indicate that SlNRC3 may contain additional amino acid polymorphisms besides K314 that contribute to its insensitivity to SS15.

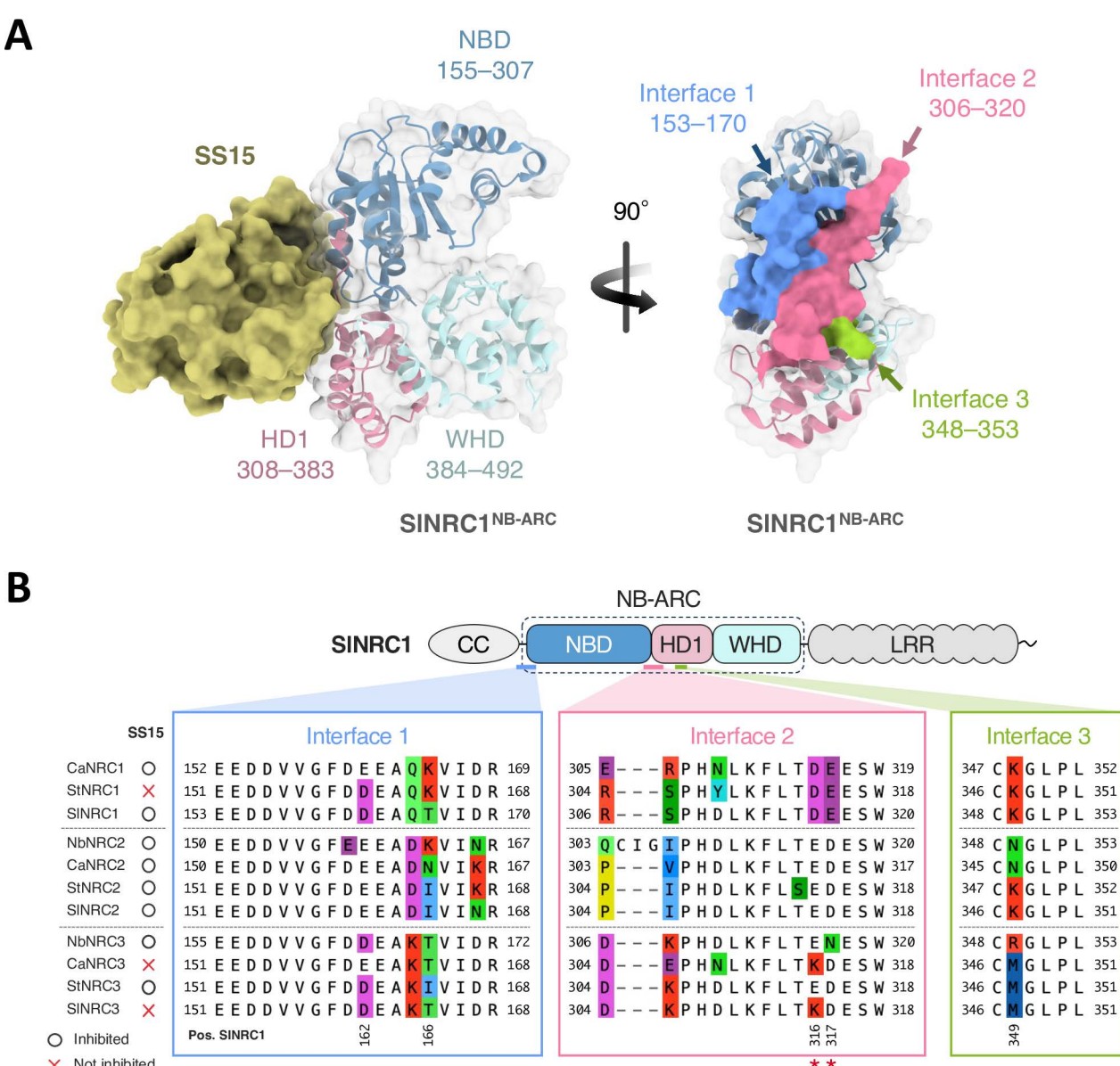

**Fig 2. NRC3 variants from different Solanaceae species are polymorphic at key residues at SS15 binding interface.** (A) Structure of SS15 in complex with the NB-ARC module of SlNRC1 (SlNRC1^NB-ARC). The NB-ARC module of SlNRC1 has three different interfaces interacting with SS15. The NB-ARC module consists of a nucleotide binding domain (NBD), a helical domain (HD1) and a winged helix domain (WHD), highlighted in different colours. The structure was published in [35] (PDB: 8BV0). The boundaries between the domains follow those in [35]. The structure is visualized using ChimeraX [43]. (B) Alignment of NRC1, NRC2 and NRC3 interfaces for SS15. The positions of the residues are based on SlNRC1. Whether an NRC is inhibited by SS15 is indicated on the right side of the protein name based on the results in Fig 1B. Two key residues for SS15 inhibition (D316 and E317 of SlNRC1), previously identified in [35], are highlighted with red asterisks.

## Polymorphisms at position E316 of NbNRC3 determine oligomerisation in the presence of SS15

Contreras et al. (2023) showed that SS15 blocks NRC2 oligomerisation and resistosome formation following activation by the NRC-S protein Rx and PVX CP [35]. We extended these experiments to NRC3 orthologs from pepper, potato and tomato to determine the effect of SS15 on their oligomerisation following activation (Fig 4). To this end, we used previously

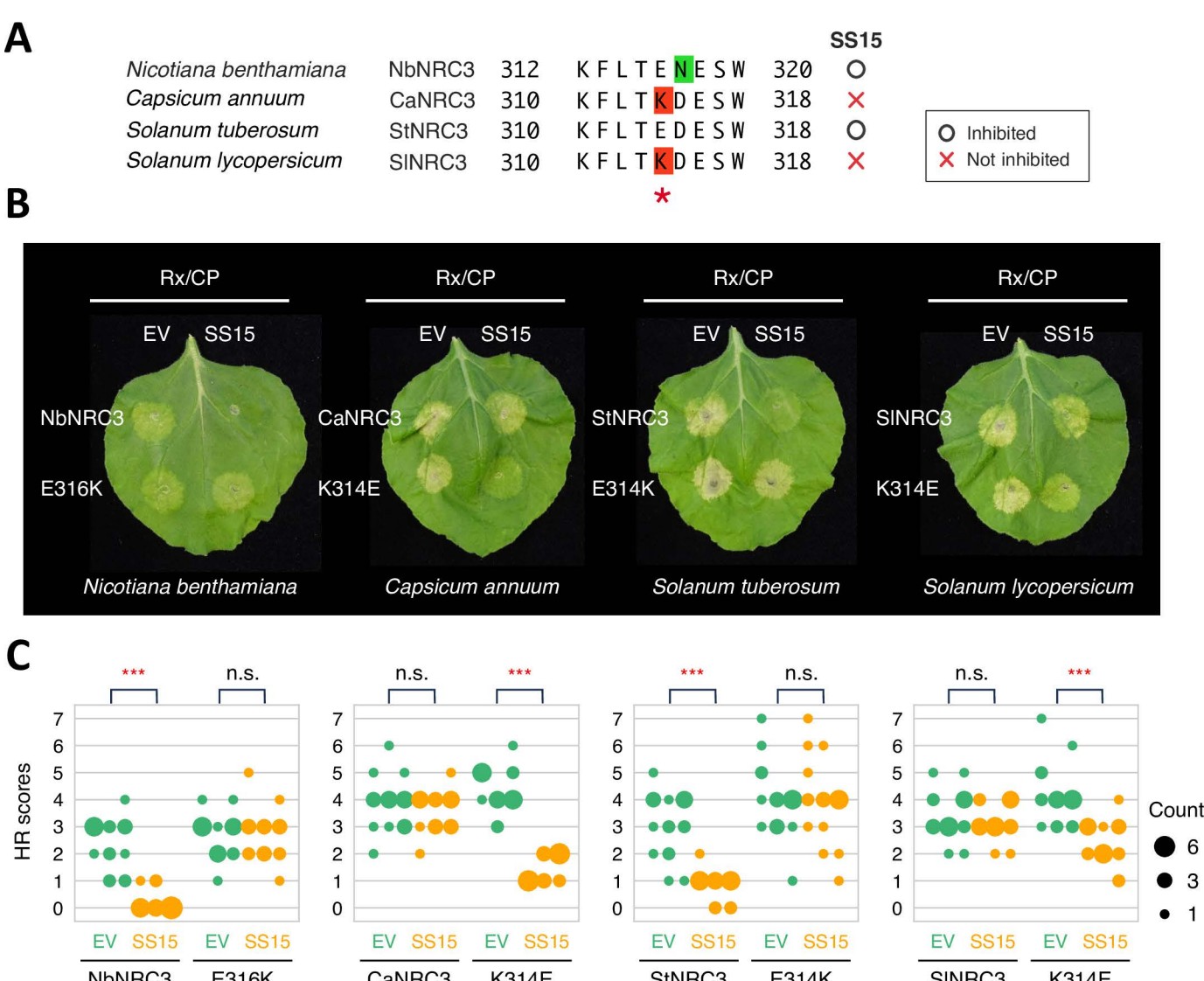

**Fig 3. Polymorphisms at NRC3 position 316 determine sensitivity to SS15 suppression.** (A) Schematic representation of the mutated sites in the multiple sequence alignment of NRC3. The residue matching D316 of SlNRC1 is highlighted with a red asterisk. Whether an NRC is inhibited by SS15 is indicated on the right side of the panel based on Fig 1B. (B) Representative images of HR cell death assays showing the results after transient co-expression of an empty vector (EV) or SS15 with Rx and PVX CP, along with either WT NRC3 or its single-point mutant in the leaves of *N. benthamiana nrc2/3/4* KO plants. Mutations were introduced in NRC3 at the residues matching D316 of SlNRC1. The leaves were photographed 5 days after infiltration. (C) Statistical analysis using a two-sided permutation test with 10,000 replicates. Statistically significant differences are indicated (***: *p* < 0.001; n.s.: not significant). Each column represents an independent experiment. The data underlying Fig 3C can be found in S1 Data.

established blue native polyacrylamide gel electrophoresis (BN-PAGE) to assay for NRC resistosome formation [33,35], and NRC3 mutants of the N-terminal MADA motifs (CaNRC3$^{EEE}$, StNRC3$^{EEE}$ and SlNRC3$^{EEE}$ for *C. annuum*, *S. tuberosum* and *S. lycopersicum*, respectively), which abolish cell death without compromising receptor activation [35,44]. To activate NRC3s, we co-expressed 4xMyc-tagged NRC3 MADA mutants (NRC3$^{EEE}$-4xMyc) with V5-tagged sensor NLR Rx (Rx-V5) and either free GFP or GFP-tagged PVX CP (CP-GFP). These effector-sensor-helper combinations were co-expressed either with mCherry-6xHA fusion protein (mCherry-6xHA) as a negative control or 4xHA-tagged SS15 (4xHA-SS15) in

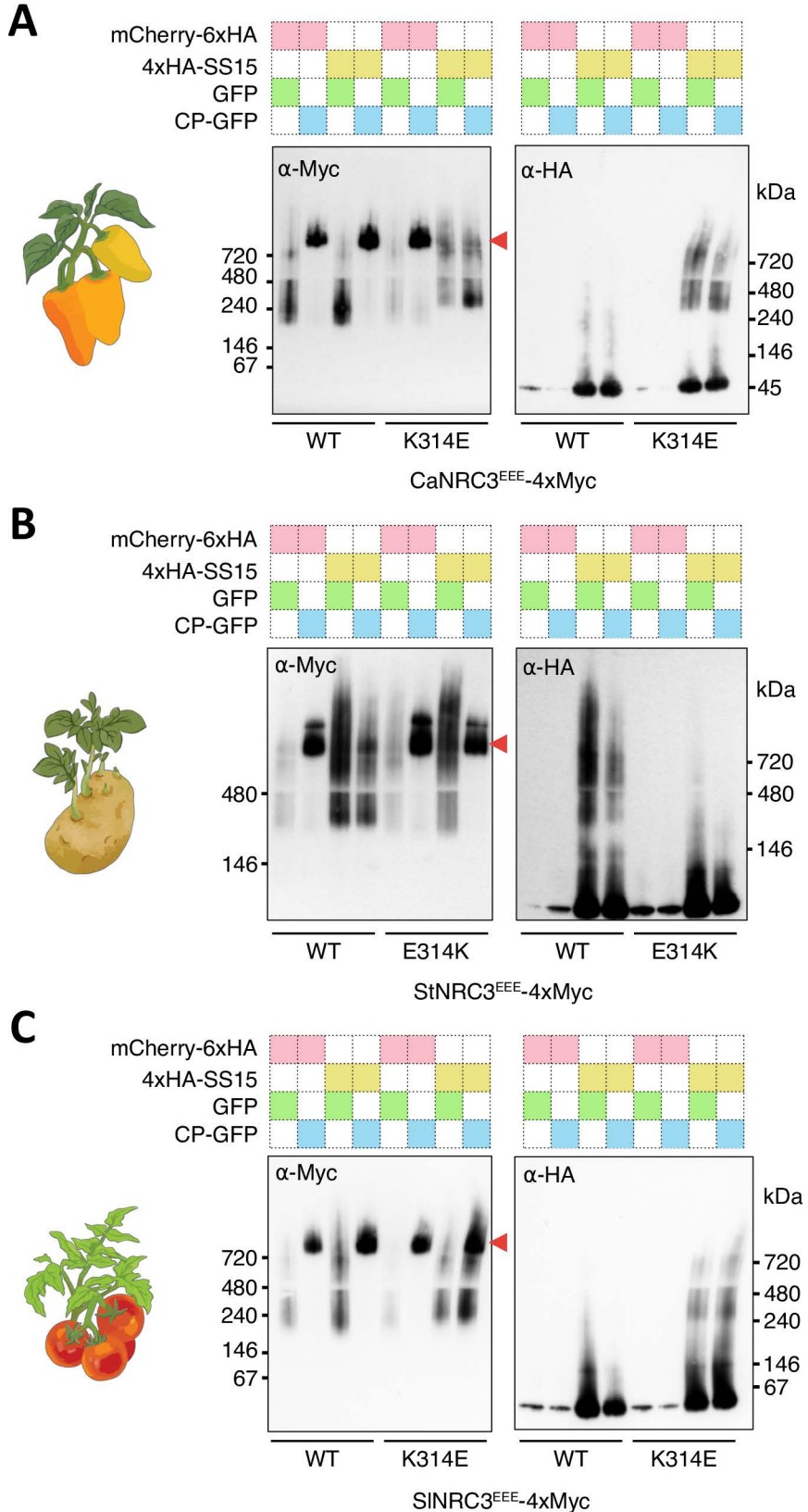

**Fig 4. Polymorphisms at position E316 of NbNRC3 determine oligomerisation in the presence of SS15.** Blue native polyacrylamide gel electrophoresis (BN-PAGE) assays were conducted for CaNRC3EEE (A), StNRC3EEE (B) and

SlNRC3$^{EEE}$ (C) with their respective single-point mutations at position matching NbNRC3 residue 316. CaNRC3$^{EEE}$, StNRC3$^{EEE}$ and SlNRC3$^{EEE}$ are the NRC3 mutants of the N-terminal MADA motifs. C-terminally 4xMyc-tagged NRC3$^{EEE}$ mutants were co-expressed with C-terminally V5-tagged Rx and either free GFP or C-terminally GFP-tagged PVX CP in the leaves of *N. benthamiana nrc2/3/4* KO plants. These effector-sensor-helper combinations were co-expressed either with mCherry-6xHA fusion protein or N-terminally 4xHA-tagged SS15. Red arrowheads indicate resistosome bands. Corresponding SDS-PAGE blots are in S4–S6 Figs.

the leaves of *N. benthamiana nrc2/3/4* KO plants (Figs 4 and S4–S6). High molecular weight bands (arrowheads in Fig 4) were visualized when WT NRC3$^{EEE}$ were co-expressed with CP, Rx and mCherry, compared with co-expression with the non-activating genes GFP and mCherry in the presence of Rx (Figs 4 and S4–S6). These results suggest that all tested WT NRC3s oligomerize and form high-order resistosome complexes upon effector-triggered activation mediated by Rx and CP. However, when NRC3$^{EEE}$s were co-expressed with SS15 in the presence of Rx and CP, the resistosome band for StNRC3$^{EEE}$ was considerably reduced in comparison to the absence of SS15, and a smear (also observed in the inactive state) appeared, showing the inhibitory effect of SS15 on StNRC3$^{EEE}$. In contrast, SS15 did not prevent resistosome formation for CaNRC3$^{EEE}$ and SlNRC3$^{EEE}$ (Figs 4 and S4–S6). Due to the low level of protein accumulation observed for NbNRC3 in *N. benthamiana*, we could not analyse the resistosome formation of NbNRC3 [41].

Next, we tested whether mutating NbNRC3 residue 316 (CaNRC3$^{EEE/K314E}$, StNRC3$^{EEE/E314K}$ and SlNRC3$^{EEE/K314E}$) affects SS15 inhibition of NRC3 oligomerization following activation as described in the previous section. Again, high molecular weight bands were visualized after co-expression of Rx and CP in BN-PAGE assays, indicating the capacity of these mutants to form a resistosome (Figs 4 and S4–S6). In contrast, the resistosome formations of CaNRC3$^{EEE/K314E}$ and SlNRC3$^{EEE/K314E}$ were impaired in the presence of SS15. In both cases, the lower molecular weight band of NRC3 exhibited a shift in size, presumably due to NRC3-SS15 complex formation *in planta* (Fig 4A and 4C). Moreover, the anti-HA blots revealed similar migration patterns for SS15 with CaNRC3$^{EEE/K314E}$ and SlNRC3$^{EEE/K314E}$, further suggesting that these mutants are forming a complex with the effector, leading to inhibition. Unlike StNRC3$^{EEE}$, the mutant StNRC3$^{EEE/E314K}$ was capable of oligomerizing in the presence of SS15 (Fig 4B). These results indicate that position E316 of NbNRC3 determines the oligomerisation of NRC3 in the presence of SS15, but in tomato SlNRC3, other residues appear to contribute to the insensitivity to SS15. Overall, these results are consistent with our observations in the HR cell death assays (Figs 1B, 3B and 3C).

## Ancestral sequence reconstruction infers that E316 is ancestral in the NRC3 clade

We revealed that polymorphisms at position 316 determine SS15 sensitivity in natural variants of NRC3. However, polymorphisms E316 of NbNRC3, K314 of CaNRC3, E314 of StNRC3 and K314 of SlNRC3 are phylogenetically discordant in the NRC3 tree (Figs 5 and S2). To clarify the evolutionary history of NRC3, we applied ancestral sequence reconstruction to 314 non-redundant NRC1/2/3/X nucleotide sequences extracted from an NLRome database of 66,665 sequences from 124 genomes in the Solanaceae family (Fig 5A and S1 Table) [42]. Ancestral sequence reconstruction inferred five amino acid substitutions from node 1 to node 6 on the three interfaces defined by the crystal structure of SlNRC1$^{NB-ARC}$ in complex with SS15 (Figs 2, 5 and S7). Node 1, which predates the divergence between the *Nicotiana* genus and the rest of the taxa, had an ambiguous residue at the position E317 of SlNRC1, resulting in two different ancestral variants anc1.1 and anc1.2 with N317 ($p = 0.45$) and D317 ($p = 0.55$),

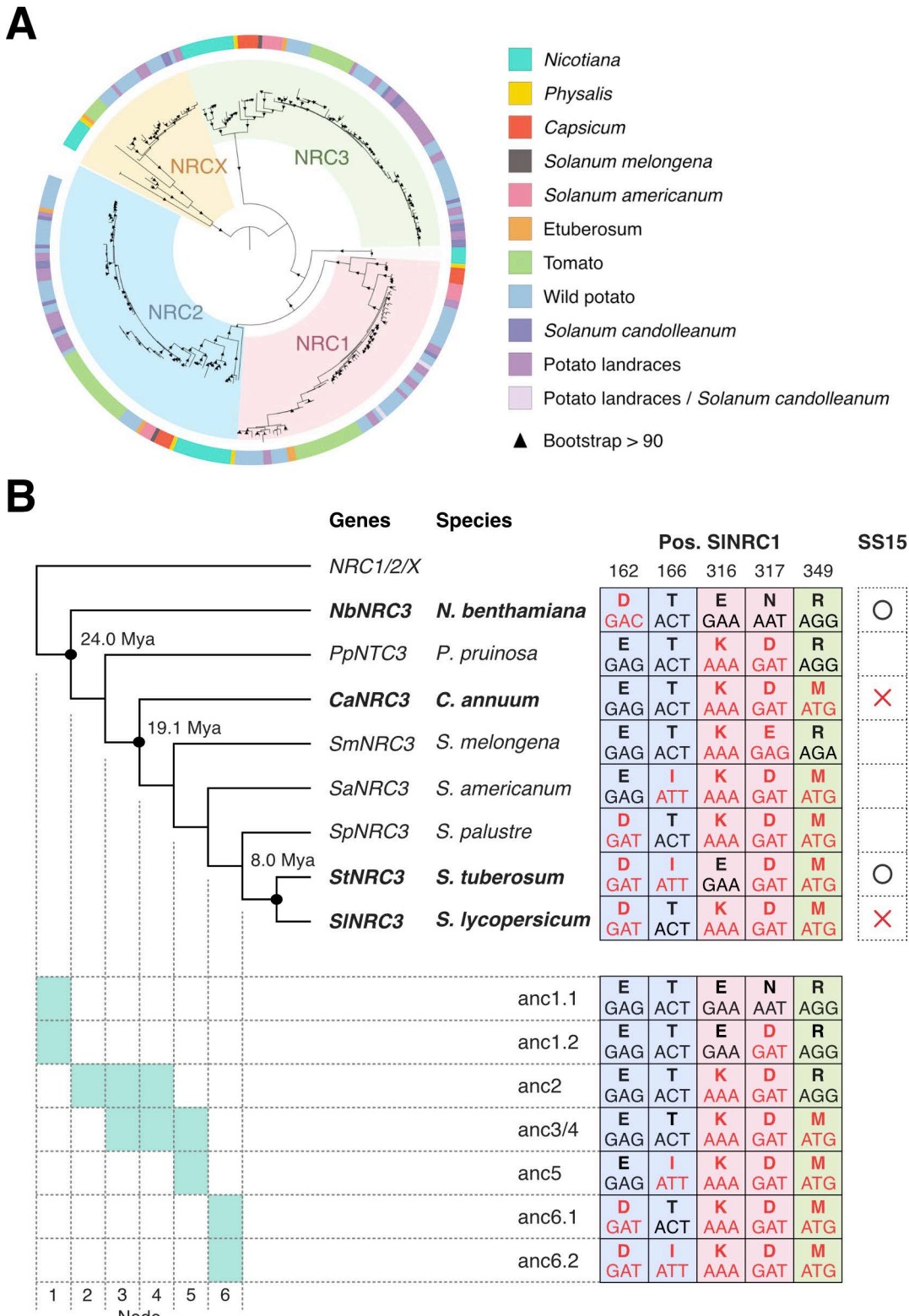

**Fig 5. Ancestral sequence reconstruction infers that E316 is ancestral in the NRC3 clade.** (A) Phylogenetic tree of 314 non-redundant nucleotide sequences from NRC1/2/3/X clade. The 314 non-redundant nucleotide sequences of the NRC1/2/3/X clade were extracted from 124 genomes in the Solanaceae family [42]. The codon-based nucleotide sequence alignment of these sequences was used to reconstruct the phylogenetic tree. The phylogenetic tree was reconstructed by the

maximum likelihood method using IQ-TREE with 1,000 bootstrap replicates. (B) Schematic illustration of the ancestral sequence reconstruction of the NRC3 clade. The ancestral sequences were inferred from a codon-based alignment of 314 sequences from the NRC1/2/3/X clade (A). We focused on the three interfaces interacting with SS15 (Fig 2), and five amino acid substitutions were identified from nodes 1 to 6. The ancestral states of remaining residues for the interfaces are identical to the amino acids of SlNRC3. The positions are based on SlNRC1 in the alignment of Fig 2B. A probability value (*p*) greater than 0.3 was considered as an ambiguous residue, and multiple ancestral states were tested from the same node (S1 Table). The correspondence of the inferred ancestral sequences with the nodes is indicated below the phylogeny. We used the calibration time estimated by [45].

respectively (Figs 5 and S7 and S1 Table). Both ancestral variants had E316 as the ancestral state, and the E316K polymorphism is inferred to have arisen during the transition from node 1–2, which are estimated to date to 24.0 (23.0–25.7) and 19.1 (17.0–21.0) million years ago (Mya) with 95% confidence intervals, respectively [45] (Fig 5B). These results indicate that the ancestral state of the NRC3 clade at position 316 was most likely E316 with the E316K polymorphism arising over ~19.1 Mya after the divergence between *Nicotiana* and the other taxa. Moreover, StNRC3 E314 is likely a regressive mutation that occurred after the divergence between *S. lycopersicum* and *S. tuberosum*.

## Ancestral NRC3 variants with E316 are inhibited by SS15, but those with K316 are not

Whereas predicted ancestral sequences at node 1 have E316 (GAA), those at node 2 to node 6 have K316 (AAA) at the key 316 position (Fig 5B). We therefore hypothesised that node 1 ancestral NRC3 is suppressed by SS15, whereas subsequent variants with K316 are not. To test this hypothesis, we swapped the SS15 binding interface of SlNRC3 (K at position 316; insensitive to SS15 inhibition) with the ancestral amino acid combinations we previously inferred (Figs 5B and 6A). Given that SlNRC3 is identical to anc6.1 at the five SS15 binding interface residues, we cloned the remainder six ancestral NRC3 sequences from node 1 to node 6 in SlNRC3 (Figs 5B and 6A), and tested them for SS15 suppression using HR and BN-PAGE assays (Figs 6 and S8–S10). These experiments revealed that SlNRC3 variants with anc1.1 and anc1.2 amino acid combinations (E316) were suppressed by SS15 for induction of HR and NRC3 oligomerisation, whereas anc2, anc3/4, anc5 and anc6.2 (K316) were not (Figs 6 and S8–S10). Overall, the difference between the NRC3 treatments with and without SS15 were clear, however, in the case of the HR assays with the anc2 variant, there was statistically significant difference between the empty vector and SS15 (S8 Fig). Nonetheless, we conclude that the anc2 variant is insensitive to SS15 given that the effector failed to suppress HR cell death and prevent resistosome formation (Figs 6 and S8–S10). These results indicate that ancestral NRC3, at the node predating the divergence between *Nicotiana* and other taxa, was inhibited by SS15. NRC3 evolved to evade SS15 inhibition over 19.1 million years ago through the E316K substitution.

To further challenge our hypothesis, we compared side-by-side anc1.1 and anc1.2 with SlNRC3 (anc6.1) and its single-point mutant (K314E) in HR cell death assays with and without SS15 (S11A and S11B Fig). SlNRC3 variants with anc1.1 and anc1.2 amino acid combinations (E316) were significantly suppressed by SS15 for induction of HR compared to SlNRC3 and the K314E mutant (S11A and S11B Fig). In the BN-PAGE assay, SS15 prevented the formation of the resistosome band for the anc1.1 variant whereas SlNRC3 and SlNRC3^K314E were insensitive to SS15 suppression (S11C and S12 Fig). These results indicate that SlNRC3 (anc6.1) possesses additional amino acid polymorphisms besides K314, which contribute to its insensitivity to SS15 (Figs 3, 6, S11 and S12).

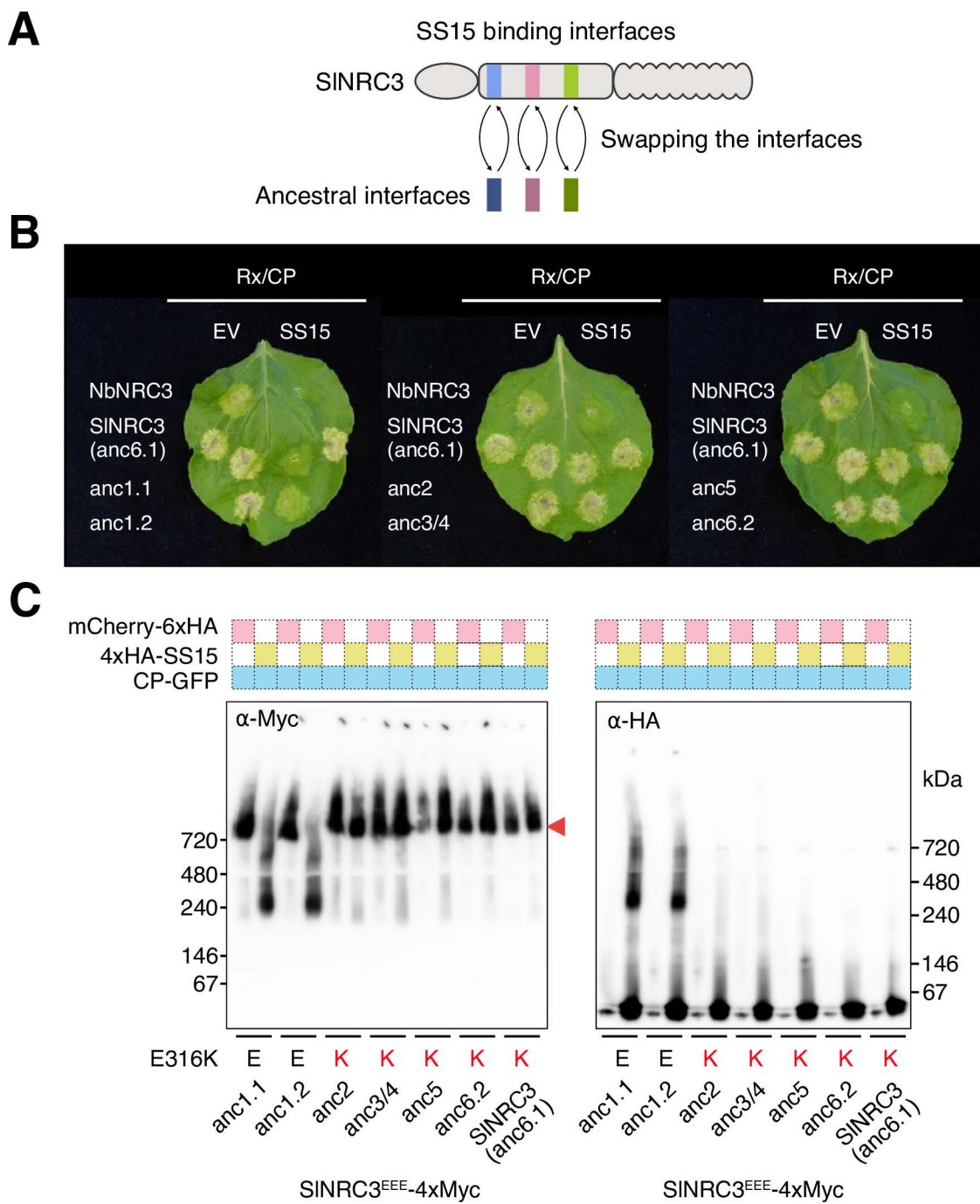

**Fig 6. Ancestral NRC3 variants with E316 are inhibited by SS15, but those with K316 are not.** (A) Overview of the strategy for resurrecting the ancestral interfaces of NRC3 for SS15. The three interfaces are defined as for SS15 in Fig 2. Using SlNRC3 as a backbone, we swapped the interface sequences with the ancestral sequences inferred in Fig 5. (B) Representative images of HR cell death assays showing the results after transient co-expression of either an empty vector (EV) or SS15 with Rx, PVX CP and the ancestral NRC3 variants in the leaves of *N. benthamiana nrc2/3/4* KO plants. NbNRC3 and SlNRC3 are used as negative and positive controls, respectively. The ancestral NRC3 variant for anc6.1 was identical to WT SlNRC3. The leaves were photographed 5 days after infiltration. The statistical analysis is in S8 Fig. (C) BN-PAGE assays for ancestral NRC3EEE variants. BN-PAGE assay was conducted for ancestral NRC3EEE variants. NRC3EEE represents the NRC3 mutant of the N-terminal MADA motif. C-terminally 4xMyc-tagged ancestral NRC3EEE variants were co-expressed with C-terminally V5-tagged Rx and C-terminally GFP-tagged PVX CP in the leaves of *N. benthamiana nrc2/3/4* KO plants. These effector-sensor-helper combinations were co-expressed either with mCherry-6xHA fusion protein or N-terminally 4xHA-tagged SS15. A red arrowhead indicates resistosome bands. Corresponding SDS-PAGE blots are in S10 Fig.

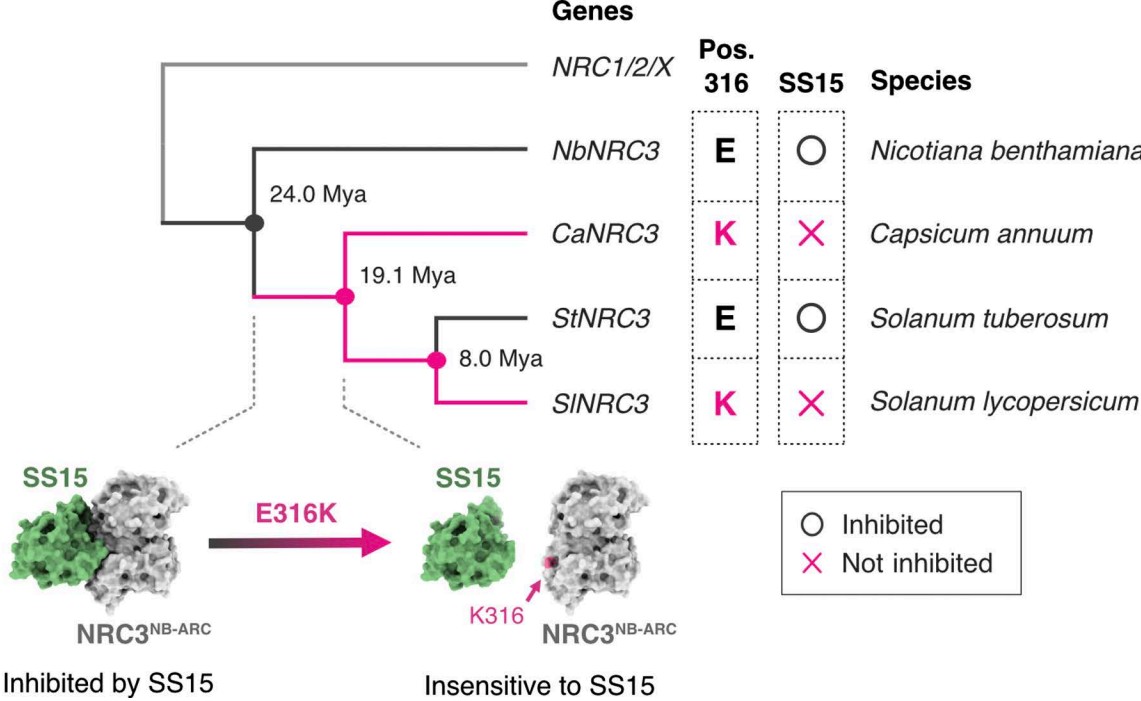

**Fig 7. NRC3 has evolved to evade inhibition by the cyst nematode effector SS15 over 19 Mya.** Polymorphisms at position E316 of NbNRC3 determine sensitivity to the cyst nematode effector SPRYSEC15 (SS15). The ancestral sequence reconstruction inferred that E316 is an ancestral state of the NRC3 clade and that E316K mutation occurred over 19 million years ago before the divergence of *Capsicum* and *Solanum* genus. The E316K mutation has made NRC3 insensitive to SS15, but NRC3 of *S. tuberosum* (StNRC3) acquired a regressive mutation (K316E) and became inhibited by SS15.

## Discussion

Plant pathogen effectors can function as suppressors of NLR-mediated immunity [15]. However, the degree to which NLRs have evolved to evade pathogen immunosuppressors has remained unknown. In this study, we discovered that some natural variants of helper NLRs in the NRC family are insensitive to suppression by the cyst nematode effector SS15. We found that some natural variants of NRC1 and NRC3 are insensitive to SS15 suppression, while all tested NRC2s are suppressed by SS15 (Fig 1). These contrasting SS15 inhibition phenotypes enabled us to determine that a single polymorphic amino acid residue at position E316 of NbNRC2 [35] determines sensitivity to SS15 in the NRC3 clade of solanaceous plants (Figs 2–4). We also tested hypotheses about adaptive evolution (from sensitive to insensitive phenotype) of the NRC-SS15 binding interface and used ancestral sequence reconstruction to show that NRC3 has evolved to evade SS15 inhibition over 19 Mya (Figs 5 and 6). These experiments led to an evolutionary model that maps out transitions in the evolution of NRC3 sensitivity to SS15 inhibition (Fig 7) and supports the view that suppressor and their receptor targets are engaged in another layer of gene-for-gene coevolution in addition to classic R-AVR interactions (S1 Fig). Intriguingly, the analyses also indicated regressive evolution (from insensitive to sensitive phenotype) in potato StNRC3 (Figs 5 and 7). Collectively, our study indicates adaptive evolution of NLRs to a pathogen immunosuppressor and identified key evolutionary transitions in a Solanaceae helper NLR that counteract this inhibition.

Our results indicate that position E316 is critical for SS15 inhibition in both NRC2 and NRC3 clades. On the other hand, the same position is not polymorphic in NRC1 sequences

across multiple species, even though they showed varying sensitivities to SS15 (Fig 1). This indicates that different residues contribute to insensitivity to SS15 in the NRC1 clade and that the sensitivity to SS15 has evolved independently in different clades of the NRC family.

Ancestral sequence reconstruction statistically infers intermediate sequences at phylogenetic nodes, enabling the identification of key evolutionary mutations. It is increasingly being applied to plant-pathogen coevolutionary systems [4,26,46–48]. For instance, this method was used to explore when the rice sensor NLR Pik-1 evolved to detect the AVR-PikD effector of the pathogenic fungus *Magnaporthe oryzae* (syn. *Pyricularia oryzae*) [47]. Similarly, ancestral sequence reconstruction was applied to the AVR-Pik effector family to understand the directionality of evolutionary changes, determining whether the effector family expanded host-target binding or host immunity evaded the effector family [46]. Moreover, a recent study employed this technique on NRC3 to examine the evolutionary history of sensor-helper compatibility [26]. In our study, we applied ancestral sequence reconstruction to NRC3 and reconstructed the evolutionary dynamics of NLRs against pathogen immunosuppressors. We analysed six evolutionary intermediate sequences of Solanaceae NRC3 to differentiate their evolutionary trajectories and discovered that SS15 sensitivity had shifted due to the E316K polymorphism that arose approximately 19 million years ago. This finding paves the way for further analyses of NLR-suppressor coevolution, particularly whether ancestral SS15 has bound ancestral NRC3 and has evolved to overcome insensitive NLRs.

Ancestral sequence reconstruction revealed the regressive evolution of potato StNRC3 to becoming sensitive to the nematode effector SS15 (Figs 5 and 7). Why did this seemingly counterintuitive evolutionary transition occur in StNRC3? One possible explanation is a trade-off between functional NLR activation and evasion of pathogen immunosuppression. In NLRs, mutations in the NB-ARC module are known to cause autoimmunity, loss-of-function, or sensor-helper incompatibility [26,49–51]. Since E316K is located in the NB-ARC module, this polymorphism could affect NLR activity, and plants carrying K316 might be subject to negative selection in the absence of parasite pressure. In this scenario, SS15 insensitivity in potato StNRC1 might be a compensating mutation that enhanced the activity of potato StNRC3 (Fig 1). Alternatively, the mutation may reflect a trade-off between SS15 evasion and another, yet unidentified, parasite or pathogen effector that interacts with StNRC3. To address this question, further investigations into the diversity of parasite or pathogen effectors that interact with NRC3 are necessary.

NRC helpers have recently been shown to form homodimers in their resting state [37]. How SS15 prevents a conversion of homodimers into resistosomes remains unclear. Importantly, the SS15-NRC interaction interface does not overlap with the NRC homodimerization interface, and co-expression of NRCs with SS15 does not result in the appearance of lower molecular weight bands of NRCs [35] (Figs 4 and 6C). This suggests that SS15 likely to binds to NRC homodimers. Obtaining structural insights into the NRC homodimer together with SS15 will shed more light on the precise molecular mechanism of SS15 inhibition.

Recent studies have reported bioengineering plant immune receptors to broaden their recognition repertoire for pathogen effectors [52–61]. An alternative bioengineering approach involves designing NLR variants that evade pathogen suppression [35,62]. A previous study utilised the differential SS15 inhibition exhibited by NbNRC2 and NbNRC4 to engineer an SS15-insensitive NRC2 variant. Our study further explored the natural diversity of NRCs within the Solanaceae family, identifying naturally occurring NRC3 variants that have evolved insensitivity to SS15 suppression (Figs 5–7). Notably, the tomato SlNRC3 harbours additional mutations at the interface, besides E314K, which confer robust insensitivity to SS15 (S11 Fig). Identifying these additional mutations could be useful to bioengineer NRC variants with enhanced robustness against SS15 inhibition. Our findings shed light on a previously

overlooked aspect of NLR-suppressor coevolution, underscoring the immune receptors' capacity to adapt to pathogen-mediated immunosuppression.

The suppressor function of the cyst nematode effector SS15 was first identified in *N. benthamiana*, whereas SS15 elicits a defence response in *Nicotiana tabacum* (tobacco), a species which also possesses the NRC network [40]. This observation suggests that SS15 may exhibit AVR activity in some host genetic backgrounds. Similarly, host-dependent AVR/suppressor activities were observed in Avr1 of *Fusarium oxysporum* f. sp. *lycopersici* infecting tomatoes [63] and AvrPtoB and AvrPphB of *Pseudomonas syringae* [64,65]. In tomato and its wild relatives, the *I* gene mediates resistance to Avr1-expressing strains of the fungus, while Avr1 suppresses the resistance mediated by *I-2* and *I-3* genes [63,66–69]. Notably, *I-2* encodes a CC-type NLR, while *I* and *I-3* encode cell-surface receptors [69]. Also, while AvrPtoB is recognised by the tomato NLR receptor complex Prf/Pto, in *Arabidopsis* it acts as a suppressor of the helper NLRs ADR1-L1/L2 [65,70,71]. Interestingly, a recent study showed that this suppression is guarded by the *Arabidopsis* sensor TIR-NLR SNC1 [65,72]. Additionally, in *Arabidopsis*, RPS5 indirectly recognises AvrPphB via PBS1 cleavage and triggers HR, while AvrPphB suppresses AvrB-induced activation of RPM1 (resistance to *P. syringae* pv. *maculicola* protein 1) [64,73]. In the case of SS15, it remains unclear how it elicits a defence response in *N. tabacum*. If an NLR recognises SS15 in *N. tabacum*, it would be interesting to determine whether this NLR binds and detects SS15 through its NB-ARC module, and how evolutionary pressure from SS15 has influenced the evolution of NLR resistance genes. These dual AVR/suppressor activities of effectors add complexity to Flor's gene-for-gene model, introducing layers of conditional coevolution between the interacting genes (S1 Fig).

Elucidating evolutionary pathways is crucial for understanding how genes and proteins have evolved to acquire their current functions [4,74,75]. In plant-pathogen interactions, various modes of gene-for-gene interactions have been proposed [14,15,26,49,76,77]. Deciphering the major evolutionary transitions underlying these gene-for-gene specificities is crucial for understanding the arms race coevolution between plants and pathogens [4,46,47]. This study leveraged natural variants of NLRs that evaded pathogen inhibitors and identified the critical evolutionary transitions that counteract these inhibitors, thereby highlighting the intricate and multilayered coevolutionary dynamics between NLRs and pathogen effectors. A better understanding of NLR-AVR and NLR-suppressor coevolution is critical to inform NLR bioengineering approaches and improve disease resistance.

## Materials and methods

### Plant growth conditions

*N. benthamiana* triple *nrc2/3/4* CRISPR knock-out (KO) mutant line '210.5.5.1' was grown in a controlled environment growth chamber with a temperature range of 22–25°C, 45–65% humidity, and a 16/8 h light/dark cycle.

### Cell death and suppression assays

Four-week-old *nrc2/3/4* KO *N. benthamiana* plants were spot-infiltrated with *Agrobacterium tumefaciens* GV3101 carrying plasmids indicated in S2 Table. Cell death, HR, was photographed 5 days post infiltration and scored according to the 0 (no necrosis) to 7 (confluent necrosis) scale (S13 Fig) as described in previous studies [41,78]. All the HR scores underlying the figures are in S1 Data. The HR scores were statistically analysed using the "permutation_test" function implemented in "SciPy" python library v1.10.0 with the options "n_resamples=10000, vectorized=False, alternative=two-sided, random_state=0". In the "permutation_test" function, we used the difference between two group means as a test statistic.

## Total protein extraction

Four-week-old *nrc2/3/4* KO *N. benthamiana* plants were infiltrated with agrobacterial suspensions as indicated in S2 Table. Leaf tissue was collected 72 h after agroinfiltration in liquid nitrogen and was grounded in a Geno/Grinder homogeniser. Proteins were extracted in GTMN buffer [10% glycerol, 50 mM tris-HCl (pH 7.5), 5 mM MgCl2, and 50 mM NaCl] supplemented with 10 mM dithiothreitol (DTT), 1x protease inhibitor cocktail (Sigma-Aldrich), and 0.2% Triton 100-X (Sigma-Aldrich), and incubated on ice for 10 min with short vortex mixing every 2 min. Debris was removed by centrifugation at 5000 *xg* for 15 min. Supernatant was used for BN-PAGE and SDS-PAGE assays.

## BN-PAGE assays

For BN-PAGE, 25 ul of extracted proteins were combined with NativePAGE 5% G-250 sample additive, NativePAGE 4x sample buffer, and water. Five ul samples were loaded and run on NativePAGE 4–16% bis-tris gels alongside SERVA Native Marker (SERVA). Proteins were transferred to PVDF membranes with NuPAGE Transfer Buffer in a Trans-Blot Turbo transfer apparatus (Bio-Rad) as per the manufacturer's instructions. Proteins were fixed to the membranes incubating with 8% acetic acid for 15 min, washed with water, and left to dry. Activation with ethanol allowed visualising the unstained native protein marker.

## SDS-PAGE assays

Extracted proteins were diluted in SDS loading dye and incubated at 72°C for 10 min and briefly centrifuged. Fifteen ul of supernatant were run on Bio-Rad 4–20% Mini-PROTEAN TGX gels with PageRuler Plus prestained protein ladder (Thermo Fisher Scientific). Proteins were transferred to PVDF difluoride membranes in Trans-Blot Turbo transfer buffer in a Trans-Blot Turbo transfer system (Bio-Rad) as per the manufacturer's instructions.

## Immunoblotting and epitope detection

BN or SDS blotted membranes were blocked with 5% milk in tris-buffered saline and 0.01% Tween 20 (TBS-T) for an hour at room temperature, and subsequently overnight at 4°C with indicated antibodies (S2 Table). Bands were visualised with Pierce ECL Western (32106, Thermo Fisher Scientific), supplementing with up to 50% SuperSignal West Femto Maximum Sensitivity Substrate (34095, Thermo Fisher Scientific) when necessary. Membranes were imaged in an ImageQuant 800 luminescent system (GE Healthcare Life Sciences, Piscataway, NJ). Rubisco loading control was stained using Ponceau S (Sigma) or Ponceau 4R (Irn Bru; AG Barr, Cumbernauld, UK).

## Interface of NRC1 and SS15

The crystal structure of tomato SlNRC1 NB-ARC module (SlNRC1[NB-ARC]) bound to SS15 effector at 4.5 Å resolution (PDB code 8BV0) was previously resolved [35]. In this structure, SS15 effector approaches the NRC1 between its NBD and HD1 domains. Molecular interactions between SlNRC1[NB-ARC] and SS15 were listed using the CONTACT program of CCP4 [79] with a distance cut-off of 2.5 to 3.6 Å. The interacting residues were clustered in three regions in SlNRC1 ranging 153–170, 306–320 and 348–353 and formed the core of the interface between SlNRC1[NB-ARC] and SS15 in the crystal structure. These three regions were named interface 1, 2 and 3, respectively. All the interfaces contain a loop region at their centre together, allowing movement of the NBD with respect to HD1 and WHD. The structure was visualized using ChimeraX [43]. To identify the polymorphisms on these interfaces, 11 NRC protein sequences

tested in HR cell death assay were aligned using MAFFT v7.508 with the options "--maxiterate 1000 --localpair" [80]. The regions corresponding to the interfaces 1, 2 and 3 were extracted from the alignment.

## Phylogenetic analyses of helper NRC clades

A previous study reported NRC helper sequences of *N. benthamiana*, *C. annuum* (pepper), *S. tuberosum* (potato) and *S. lycopersicum* (tomato) [37]. The full-length amino acid sequences of the NRC0 (5 sequences), NRC1 (6 sequences), NRC2 (10 sequences), NRC3 (7 sequences) and NRCX (3 sequences) were obtained from this previously published dataset. Exact matches for the NbNRC2, CaNRC3 and StNRC3 sequences used in the HR cell death assay were not found within the dataset. Therefore, they were included in the dataset. The full-length amino acid sequences of helper NRC0/1/2/3/X clades were aligned using MAFFT v7.508 with the options "--maxiterate 1000 --localpair" [80]. This alignment was then trimmed using ClipKIT v1.3.0 with the options "-c -m gappy -g 0.9" [81]. Based on the trimmed alignment, a phylogenetic tree was created using IQ-TREE v1.6.12 [82] with 1,000 ultrafast bootstrap replicates [83]. ModelFinder selected the best-fit model for tree reconstruction, "JTT+G4", according to the Bayesian Information Criterion (BIC) [84]. The resulting tree was visualised using Iroki [85], and the bootstrap values for key nodes were shown in the figures.

## Obtaining codon-based alignment for ancestral sequence reconstruction

Sequences of 2,361 helper NRCs were obtained from a previously published dataset [26]. The dataset was cleaned by removing sequences containing internal stop codons, those lacking a multiple-of-three nucleotide length, and terminal stop codons, resulting in 2,353 sequences. Then, sequences shorter than 2,400 or longer than 2,800 bases were filtered out, resulting in 1,748 sequences. To identify the NRC1/2/3/X clade, the protein sequences of the NB-ARC module were aligned using MAFFT v7.508 with the options "--maxiterate 1000 --localpair" [80]. The alignment was trimmed using ClipKIT v1.3.0 with the options "-c -m gappy -g 0.9" [81]. Based on the trimmed alignment, a phylogenetic tree was created using IQ-TREE v2.2.6 [86] with 1,000 ultrafast bootstrap replicates [83]. ModelFinder selected the best-fit model for tree reconstruction, "JTT+F+R6", according to the BIC [84]. Based on this phylogenetic tree, the 591 full-length nucleotide sequences were obtained from the NRC1/2/3/X clade. These 591 sequences were translated using seqkit v2.3.0 "translate" command [87] and then re-aligned using MAFFT v7.508 with the options "--maxiterate 1000 --localpair" [80]. The position T4 of "Ppr-t_12226" was manually edited in the alignment. To obtain a codon-based nucleotide sequence alignment, the nucleotide sequences were threaded onto the protein alignment using the "thread_dna" command in phykit v1.11.14 [88]. The alignment was trimmed using ClipKIT v1.3.0 with the options "-c -m gappy -g 0.9" [81]. Finally, duplicated sequences were removed using seqkit v2.3.0 "rmdup" command [87] with the -s and -P options. Some duplicated splicing variants were also filtered out, resulting in 314 non-redundant nucleotide sequences.

## Ancestral sequence reconstruction

Based on the codon-based nucleotide alignment, which contains 314 non-redundant nucleotide sequences from the NRC1/2/3/X clade, a phylogenetic tree was created using IQ-TREE v2.2.6 [86] with 1,000 ultrafast bootstrap replicates [83]. ModelFinder selected the best-fit model for tree reconstruction, "TIM3+F+I+R4", according to the BIC [84]. An empirical Bayes method implemented in IQ-TREE was used to infer the ancestral states of the input sequences [86,89,90]. The previously generated alignment and tree files were input to IQ-TREE, specifying the options "-s DNA -o SlNRCX -m TIM3+F+I+R4". Since IQ-TREE does not infer the

ancestral states of insertion and deletion events (indels), the sequence alignment was converted to a simple binary presence-absence alignment file. In this file, "0" and "1" represent non-gap and gap, respectively. The Jukes-Cantor type binary model (JC2) in IQ-TREE was used to infer the ancestral states of this binary data, similar to how a previous study did with RAxML [91,92]. For indel reconstruction, the branch lengths of the tree were fixed with the "-blfix" option. Finally, the results of ancestral sequence reconstruction were combined with those of ancestral indel reconstruction, and the probabilities of each codon were calculated by simply multiplying the probabilities of each nucleotide. A probability value ($p$) greater than 0.3 was considered as an ambiguous residue. Focusing on the three interfaces defined by the crystal structure of SlNRC1$^{NB-ARC}$ in complex with SS15, six different ancestral variants (anc1.1, anc1.2, anc2, anc3/4, anc5, anc6.2) were cloned and tested (S2 Table). The ancestral interfaces of anc6.1 were identical to those of WT SlNRC3. This programming pipeline was named "ancseq" and deposited on GitHub (https://github.com/YuSugihara/ancseq).

### Consensus sequence analyses

The NRC sequences and alignment in Fig 5A were used as input. The alignments were converted into a matrix using the "alignment_to_matrix" function in logomaker (Tareen and Kinney, 2020) with the options (to_type='information', pseudocount=0), and the logo plots were generated with the "weblogo_protein" color scheme in logomaker. The NRC1/2/3 clade was distinguished from the NRCX clade, where four sequences were not clearly assigned to either the NRC1, NRC2, or NRC3 clades (Fig 5A).

## Supporting information

**S1 Fig. Multiple levels of gene-for-gene interactions between pathogens and plants.** To date, most studies have focused on coevolution between R (immune receptors or *R* genes) and AVR effectors, of which there are numerous examples. On the other hand, coevolution between pathogen effectors with immunosuppression activities (suppressors) and their receptor targets are less understood.
(TIF)

**S2 Fig. Phylogenetic tree of helper NRC0/1/2/3/X clades and sequences used in Fig 1B.** We used the NRC sequences of the NRC0, NRC1, NRC2, NRC3 and NRCX clades from four different Solanaceae species (*Nicotiana benthamiana*, *Capsicum annuum*, *Solanum tuberosum* and *Solanum lycopersicum*) and created a phylogenetic tree. When a cloned sequence is not identical to any sequences in the dataset, we added them to the dataset. Each cloned NRC sequence is highlighted in red. When a cloned sequence is identical to multiple NRC sequences in the tree, they are indicated in the figure. The phylogenetic tree is reconstructed by the maximum likelihood method using IQ-TREE with 1,000 bootstrap replicates.
(TIF)

**S3 Fig. Protein accumulations of NRC3s and their single-point mutants in the presence of SS15.** C-terminally 4xMyc-tagged NRC3s and single-point mutants were expressed with N-terminally StrepII-tagged SS15 in the leaves of *N. benthamiana nrc2/3/4* KO plants. NbNRC3 expressed with EV was used as a negative control for the anti-Strep blot. Rubisco loading control was carried out using Ponceau staining (PS).
(TIF)

**S4 Fig. SDS-PAGE accompanying BN-PAGE shown in Fig 4A.** SDS-PAGE assay was conducted for CaNRC3$^{EEE}$ and its single-point mutant at position matching NbNRC3 residue 316.

CaNRC3[EEE] represents the CaNRC3 mutant of the N-terminal MADA motif. C-terminally 4xMyc-tagged CaNRC3[EEE] mutants were co-expressed with C-terminally V5-tagged Rx and either free GFP or C-terminally GFP-tagged PVX CP in the leaves of *N. benthamiana nrc2/3/4* KO plants. These effector-sensor-helper combinations were co-expressed either with mCherry-6xHA fusion protein or N-terminally 4xHA-tagged SS15. Rubisco loading control was carried out using Ponceau staining (PS).
(TIF)

**S5 Fig. SDS-PAGE accompanying BN-PAGE shown in Fig 4B.** SDS-PAGE assay was conducted for StNRC3[EEE] and its single-point mutant at position matching NbNRC3 residue 316. StNRC3[EEE] represents the StNRC3 mutant of the N-terminal MADA motif. C-terminally 4xMyc-tagged StNRC3[EEE] mutants were co-expressed with C-terminally V5-tagged Rx and either free GFP or C-terminally GFP-tagged PVX CP in the leaves of *N. benthamiana nrc2/3/4* KO plants. These effector-sensor-helper combinations were co-expressed either with mCherry-6xHA fusion protein or N-terminally 4xHA-tagged SS15. Rubisco loading control was carried out using Ponceau staining (PS).
(TIF)

**S6 Fig. SDS-PAGE accompanying BN-PAGE shown in Fig 4C.** SDS-PAGE assay was conducted for SlNRC3[EEE] and its single-point mutant at position matching NbNRC3 residue 316. SlNRC3[EEE] represents the SlNRC3 mutant of the N-terminal MADA motif. C-terminally 4xMyc-tagged SlNRC3[EEE] mutants were co-expressed with C-terminally V5-tagged Rx and either free GFP or C-terminally GFP-tagged PVX CP in the leaves of *N. benthamiana nrc2/3/4* KO plants. These effector-sensor-helper combinations were co-expressed either with mCherry-6xHA fusion protein or N-terminally 4xHA-tagged SS15. Rubisco loading control was carried out using Ponceau staining (PS).
(TIF)

**S7 Fig. Consensus sequence patterns for SS15 interface regions of each NRC clade.** The consensus sequence patterns of each NRC clade and the NRC1/2/3 clade were generated using logomaker [93]. We used the NRC sequences and alignment from Fig 5A. The NRC1/2/3 clade was distinguished from the NRCX clade, where four sequences were not clearly assigned to either the NRC1, NRC2, or NRC3 clade (Fig 5A).
(TIF)

**S8 Fig. Statistical analysis of Fig 6B.** We used a two-sided permutation test with 10,000 replicates. Statistically significant differences are indicated (***: $p < 0.001$; n.s.: not significant). Each column represents an independent experiment. The data underlying this figure can be found in S1 Data.
(TIF)

**S9 Fig. Protein accumulations of ancestral NRC3 variants in the presence of SS15.** C-terminally 4xMyc-tagged ancestral NRC3 variants were expressed with N-terminally StrepII-tagged SS15 in the leaves of *N. benthamiana nrc2/3/4* KO plants. The ancestral NRC3 variant for anc6.1 was identical to WT SlNRC3 at the five SS15 binding interface residues. NbNRC3 expressed with EV was used as a negative control for the anti-Strep blot. Rubisco loading control was carried out using Ponceau staining (PS).
(TIF)

**S10 Fig. SDS-PAGE accompanying BN-PAGE shown in Fig 6C.** SDS-PAGE assay was conducted for ancestral NRC3[EEE] variants. NRC3[EEE] represents the NRC3 mutant of the N-terminal MADA motif. C-terminally 4xMyc-tagged ancestral NRC3[EEE] variants were

co-expressed with C-terminally V5-tagged Rx and C-terminally GFP-tagged PVX CP in the leaves of *N. benthamiana nrc2/3/4* KO plants. These effector-sensor-helper combinations were co-expressed either with mCherry-6xHA fusion protein or N-terminally 4xHA-tagged SS15.
(TIF)

**S11 Fig.  Tomato NRC3 accumulates mutations that confer robust insensitivity to SS15.**
(A) Representative images of HR cell death assays showing the results after transient co-expression of either an empty vector (EV) or SS15 with Rx and PVX CP, along with either WT SlNRC3, SlNRC3$^{K314E}$ or the ancestral NRC3 variants (anc1.1 and anc1.2) in the leaves of *N. benthamiana nrc2/3/4* KO plants. The anc1.1 and anc1.2 are the ancestral NRC3 variants tested in Fig 6. The leaves were photographed 5 days after infiltration. (B) Statistical analysis of S11A Fig using a two-sided permutation test with 10,000 replicates. Statistically significant differences are indicated (***: $p < 0.001$; n.s.: not significant). Each column represents an independent experiment. (C) BN-PAGE assays for WT SlNRC3, SlNRC3$^{K314E}$ and anc1.1. The SlNRC3$^{EEE}$s, the N-terminal MADA motif mutants, were used in this BN-PAGE assay. C-terminally 4xMyc-tagged NRC3s were co-expressed with C-terminally V5-tagged Rx and C-terminally GFP-tagged PVX CP in the leaves of *N. benthamiana nrc2/3/4* KO plants. These effector-sensor-helper combinations were co-expressed either with mCherry-6xHA fusion protein or N-terminally 4xHA-tagged SS15. A red arrowhead indicates resistosome bands. Corresponding SDS-PAGE blots are in S12 Fig. The data underlying S11B Fig can be found in S1 Data.
(TIF)

**S12 Fig.  SDS-PAGE accompanying BN-PAGE shown in S11C Fig.** SDS-PAGE assays were conducted for WT SlNRC3, SlNRC3$^{K314E}$ and anc1.1. The SlNRC3$^{EEE}$s, the N-terminal MADA motif mutants, were used in this SDS-PAGE assay. C-terminally 4xMyc-tagged NRC3 variants were co-expressed with C-terminally V5-tagged Rx and either free GFP or C-terminally GFP-tagged PVX CP in the leaves of *N. benthamiana nrc2/3/4* KO plants. These effector-sensor-helper combinations were co-expressed either with mCherry-6xHA fusion protein or N-terminally 4xHA-tagged SS15. Rubisco loading control was carried out using Ponceau staining (PS).
(TIF)

**S13 Fig.  HR scale used in this study.** Cell death, HR, was photographed 5 days post infiltration and scored according to the 0 (no necrosis) to 7 (confluent necrosis) scale as described in previous studies [41,78].
(TIF)

**S1 Table.  Summary of ancestral sequence reconstruction probabilities.**
(XLSX)

**S2 Table.  Cloning details of constructs used in this study.**
(XLSX)

**S1 Data.  Underlying numerical data for Figs 3C, S8 and S11B.**
(XLSX)

## Acknowledgements

C.M.-A. is grateful to the DGAPA-PASPA UNAM Program for financing a sabbatical year at TSL. We thank all members of the TSL Support Services for their invaluable assistance.

## Author contributions

**Conceptualization:** Jiorgos Kourelis, Sophien Kamoun.

**Data curation:** Yu Sugihara, Jiorgos Kourelis, AmirAli Toghani, Claudia Martínez-Anaya.

**Formal analysis:** Yu Sugihara, Muniyandi Selvaraj, Claudia Martínez-Anaya.

**Funding acquisition:** Sophien Kamoun.

**Investigation:** Yu Sugihara, Jiorgos Kourelis, Claudia Martínez-Anaya.

**Methodology:** Yu Sugihara, Mauricio P. Contreras, Hsuan Pai.

**Project administration:** Sophien Kamoun.

**Resources:** Jiorgos Kourelis.

**Software:** Yu Sugihara.

**Supervision:** Jiorgos Kourelis, Mauricio P. Contreras, Claudia Martínez-Anaya, Sophien Kamoun.

**Validation:** Yu Sugihara, Mauricio P. Contreras, Adeline Harant, Claudia Martínez-Anaya.

**Visualization:** Yu Sugihara, Mauricio P. Contreras, Hsuan Pai, Muniyandi Selvaraj, AmirAli Toghani, Claudia Martínez-Anaya, Sophien Kamoun.

**Writing – original draft:** Yu Sugihara, Jiorgos Kourelis, Mauricio P. Contreras, Muniyandi Selvaraj, Claudia Martínez-Anaya, Sophien Kamoun.

**Writing – review & editing:** Yu Sugihara, Jiorgos Kourelis, Mauricio P. Contreras, Claudia Martínez-Anaya, Sophien Kamoun.

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
