## [Decision Letter · Decision Letter 0]

11 Oct 2024

Dear Dr Kamoun,

Thank you very much for submitting your Research Article entitled 'Helper NLR immune protein NRC3 evolved to evade inhibition by a cyst nematode virulence effector' to PLOS Genetics.

The manuscript was fully evaluated at the editorial level and by independent peer reviewers. The reviewers appreciated the attention to an important problem, but raised some substantial concerns about the current manuscript. Based on the reviews, we will not be able to accept this version of the manuscript, but we would be willing to review a much-revised version. We cannot, of course, promise publication at that time.

If you decide to revise the manuscript for further consideration at PLOS Genetics, please aim to resubmit within the next 60 days, unless it will take extra time to address the concerns of the reviewers, in which case we would appreciate an expected resubmission date by email to plosgenetics@plos.org.

If present, accompanying reviewer attachments are included with this email; please notify the journal office if any appear to be missing. They will also be available for download from the link below. You can use this link to log into the system when you are ready to submit a revised version, having first consulted our Submission Checklist .

PLOS has incorporated Similarity Check , powered by iThenticate, into its journal-wide submission system in order to screen submitted content for originality before publication. Each PLOS journal undertakes screening on a proportion of submitted articles. You will be contacted if needed following the screening process.

To resubmit, log into your Editorial Manager account and select the option 'Revise Submission' in the 'Submissions Needing Revision' folder.

We are sorry that we cannot be more positive about your manuscript at this stage. Please do not hesitate to contact us if you have any concerns or questions.

Yours sincerely,

Tiancong Qi

Academic Editor

PLOS Genetics

Claudia Köhler

Section Editor

PLOS Genetics

Reviewer's Responses to Questions

**Comments to the Authors:**

Reviewer #1: In this study, while the cyst nematode virulence effector SPRYSEC15 (SS15) binds and inhibits oligomerisation of helper NLR proteins, the authors found some natural variants of NRC1 and NRC3 are insensitive to SS15 suppression. Further, they claimed that polymorphisms at NRC3 position 316 determine sensitivity to SS15 suppression by influencing NRC3-SS15 interaction. Additionally, ancestral sequence reconstruction revealed that NRC3 transitioned from an ancestral suppressed form to an insensitive one over 19 million years ago, suggesting the intricate and multilayered coevolutionary dynamics between NLRs and pathogen effectors. The concept of the story is intriguing; however, some important experiments are missing and should be supplemented.

1. The protein level of NRC3 and mutants in all HR cell death assays should be tested to exclude the instability or expression differences of mutants co-expression with SS15.

2. The authors claimed E316K substitution on NRC3 influence NRC3-SS15 binding, thereby influencing SS15 inhibition of NRC3 mediated HR cell death. However, the data of SS15 interaction with NRC3 or variants were missing.

3. Since NRC3 is a core helper NLR required not only by sensor NLRs but also by cell surface receptors, the subcellular localization of NRC3 & mutants is critical to their function and should be examined.

4. In their study, the effects of single-point mutants (E316K in NbNRC3, K314E in CaNRC3, E314K in StNRC3) are remarkable in HR cell death assays and BN-PAGE assays (Fig2B, Fig4, Fig6B-C), suggesting position 316 (K or E) determines NRC3 oligomerisation and sensitivity to SS15 inhibition. Since Contreras et al. (2023) described the three contact interfaces in the SlNRC1NB-ARC complex. How could one amino acid residue plays such the dominant role?

Reviewer #2: I previously reviewed the manuscript for a different publication and had an overall positive opinion, which holds true for this submission. I provide the same comments below, with edits after reviewing the present version.

The NRC sensor-helper network is a well characterized set of plant NLR receptors. Diverse sensor NRCs are related to, and genetically dependent on, a more conserved clade of helper NRCs, and the proteins make up an immunity module which has evolved many different pathogen recognition functions in Solanaceous plants. Recent landmark studies showed that a nematode efector SS15 and an oomycete effector can suppress immune responses through interaction with helper NRCs, and in the case of SS15 the interaction blocks the formation of the oligomeric helper NRC3 resistosome.

The present manuscript describes SS15 specificity against a broad range of helper NRC homologs (beyond the previously studied N.benthamiana paralogs). Naturally occuring SS15-evasive NRC3 homologs show HR insensitivity and BN-page oligomer resistance to SS15 perturbation. Through extensive phylogenomic analysis and functional testing of point mutants, the manuscript further shows that a specific NRC3 NBARC interface evolved the SS15-evasive mutation in the ancestor of Physalis / Capsicum / Solanum genera (E316K), but has undergone regressive evolution in potato StNRC3 (K316E) to again become senstive to SS15. Ancestrally reconstructed sequences at the interface are swapped into the SlNRC3 backbone to elegantly show a series of SS15 inhibition and evasion events over an inferred evolutionary timeline.

Overall the work is of high quality. Results are very clearly presented. The findings are impactful for several reasons. First, a broad survey of NBARC sequence variation and evolution motivates interest across all plant and animal NLRs at homologous HD1 subdomain interfaces, since the surface may serve as a target for other effectors to suppress oligomerization. Second, combining phylogenomic inference with structural models (empirical or predicted) is also an timely, powerful pipeline. The manuscript describes an excellent github repository "ancseq" to help other researchers use the pipeline. As usual for the research group, supplemental figures and supporting Zenodo datasets are easily accessible. I have 5 relatively minor comments.

1) A limitation of the study is that only a single extant SS15 sequence is used from yellow potato cyst nematode, Globodera rostochiensis. The ancestral SS15 variants which may have formed interfaces with NRC3 ancestors are not explored.

The manuscript could add data on extant SS15 sequence diversity, and whether an ancestral SS15 was likely to have similar specificity against NRC3. At least two Globodera rostochiensis genomes are available at DOI: 10.1094/PHYTO-09-20-0403-A (raw reads could be searched by SRA blast), and related Globodera also have sequenced genomes http://nemaplex.ucdavis.edu/Taxadata/G053.aspx.

2) The results imply that there is pressure on the key "toggling" residue 316 to revert to E even if the evasive K316 substitution exists. A discussion paragraph hypothesizes that K316 may confer a degree of autoactivity or autoimmunity, driving regression to E in the absence of the pathogen. If this is the case, would one predict that K316 variants are "trigger happy"? To test this, an experiment using lower levels of Rx/CP (lower OD of agroinfiltration) might be sufficient to activate similar levels of HR when K316 NRC3 variants are expresed, compared to E316. This experiment, or another demonstration of an advantage of E316, would help to explain the conflicting evolutionary pressures. An alternative hypothesis is that arms race dynamics with a specific electrostatic interaction with SS15 are driving reversion.

3) The reconstructed sequences of nodes 1-6 in Fig 5 were inferred from a larger alignment of 314 sequences of the NRC1/2/3/X clade (Fig. S3). Rather than presenting a schematic diagram of just 8 homologs, it would be better to move Fig. S3 to the main text to show the true number of NRC3 homologs used for inference.

4) To visualize sequence conservation and diversity for NRC1/2/3 clades for the entire 314 sequence dataset, sequence logos for interfaces 1-3 could be added to Fig. 2. (Similar to Fig. 4B of Selvaraj et al 2023)

5) Line 314: What is the overall sequence similarity of ancestrally reconstructed sequences to extant NRC3? An alignment figure of all ancestral sequences could be added as a supplemental figure.

Reviewer #3: Attached review file

**Have all data underlying the figures and results presented in the manuscript been provided?**

Reviewer #1: None

Reviewer #2: Yes

Reviewer #3: Yes

PLOS authors have the option to publish the peer review history of their article (what does this mean? ). If published, this will include your full peer review and any attached files.

**Do you want your identity to be public for this peer review?** For information about this choice, including consent withdrawal, please see our Privacy Policy .

Reviewer #1: No

Reviewer #2: No

Reviewer #3: **Yes: ** Adi Avni

---

## [Decision Letter · Decision Letter 1]

9 Mar 2025

Dear Dr Kamoun,

We are pleased to inform you that your manuscript entitled "Helper NLR immune protein NRC3 evolved to evade inhibition by a cyst nematode virulence effector" has been editorially accepted for publication in PLOS Genetics. Congratulations!

Yours sincerely,

Tiancong Qi

Academic Editor

PLOS Genetics

Aimée Dudley

Editor-in-Chief

PLOS Genetics

Aimée Dudley

Editor-in-Chief

PLOS Genetics

Anne Goriely

Editor-in-Chief

PLOS Genetics

Comments from the reviewers (if applicable):

Reviewer's Responses to Questions

**Comments to the Authors:**

Reviewer #1: The authors have addressed the majority of my comments. With respect to experiments, I have no further concerns.

Reviewer #2: The revised manuscript has made edits which clarify the findings. The larger NRC phylogenetic tree has been moved to main Fig. 5. Sequence logos help portray the interfaces in terms of both variable and conserved AAs. I am satisfied with the revision and hope it drives interest in mining natural variation at host target interfaces.

Reviewer #3: I am reviewing this manuscript for the second time. The authors answered all my questions and remarks.

**Have all data underlying the figures and results presented in the manuscript been provided?**

Reviewer #1: Yes

Reviewer #2: Yes

Reviewer #3: Yes

PLOS authors have the option to publish the peer review history of their article (what does this mean? ). If published, this will include your full peer review and any attached files.

**Do you want your identity to be public for this peer review?** For information about this choice, including consent withdrawal, please see our Privacy Policy .

Reviewer #1: **Yes: ** Yunjing Wang

Reviewer #2: No

Reviewer #3: No

**Data Deposition**

http://datadryad.org/submit?journalID=pgenetics&manu=PGENETICS-D-24-01047R1

**Press Queries**

---

## [Editor Report · Acceptance letter]

PGENETICS-D-24-01047R1

Helper NLR immune protein NRC3 evolved to evade inhibition by a cyst nematode virulence effector

Dear Dr Kamoun,

We are pleased to inform you that your manuscript entitled "Helper NLR immune protein NRC3 evolved to evade inhibition by a cyst nematode virulence effector" has been formally accepted for publication in PLOS Genetics! Your manuscript is now with our production department and you will be notified of the publication date in due course.

With kind regards,

Anita Estes

PLOS Genetics

On behalf of:
